

# Potential impacts of finfish aquaculture on eelgrass (*Zostera marina*) beds and possible monitoring metrics for management: a case study in Atlantic Canada

Nakia Cullain, Reba McIver, Allison L. Schmidt, Inka Milewski and Heike K. Lotze

Department of Biology, Dalhousie University, Halifax, Canada

## ABSTRACT

Eelgrass (*Zostera marina*) has been designated an Ecologically Significant Species in Atlantic Canada. The development and rapid expansion of netpen finfish aquaculture into sensitive coastal habitats has raised concerns about the impacts of finfish aquaculture on eelgrass habitats. To date, no studies have been done in Atlantic Canada to examine these impacts or to identify potential monitoring variables that would aid in the development of specific conservation and management objectives. As a first step in addressing this gap, we examined differences in environmental variables, eelgrass bed structure and macroinfauna communities at increasing distances from a finfish farm in Port Mouton Bay, a reference site in adjacent Port Joli Bay, and published survey results from other sites without finfish farms along the Atlantic Coast of Nova Scotia. Drawing on research done elsewhere and our results, we then identified possible metrics for assessing and monitoring local impacts of finfish aquaculture on eelgrass habitats. Our results suggest some nutrient and organic enrichment, higher epiphyte loads, lower eelgrass cover and biomass, and lower macroinfauna biomass closer to the farm. Moreover, community structure significantly differed between sites with some species increasing and others decreasing closer to the farm. Changes in the macroinfauna community could be linked to observed differences in environmental and eelgrass bed variables. These results provide new insights into the potential impacts of finfish aquaculture on eelgrass habitats in Atlantic Canada. We recommend a suite of measures for assessment and monitoring that take into account response time to disturbance and account for different levels of eelgrass organizational response (from physiological to community).

# INTRODUCTION

Seagrass beds are among the most productive and diverse ecosystems on the planet (*Larkum, Orth & Duarte, 2006*); however, they continue to be threatened by anthropogenic activities

Corresponding author
Nakia Cullain, nakia@zavoralab.com

around the world (*Orth et al., 2006*; *Waycott et al., 2009*; *Short et al., 2011*). Discharge of nutrients and accumulation of organic matter from human sources, such as municipal and industrial wastewater and land run-off are some of the most influential causes of degradation to seagrass habitats in coastal waters (*Hauxwell, Cebrian & Valiela, 2003*; *Waycott et al., 2009*; *Schmidt et al., 2012*). Over the past two decades, the development and rapid expansion of finfish aquaculture (such as open netpens) into sensitive coastal habitats has further increased the risk of degradation to seagrasses (*Duarte, 2002*; *Vandermeulen, 2005*; *Holmer et al., 2008*).

The impacts of marine fish farms on seagrass beds have been examined extensively, but not exclusively, on *Posidonia oceanica* meadows in the Mediterranean Sea (Table 1). These studies have shown that increased nutrient loading, organic matter and sedimentation rates resulting from fish farm activities can lead to changes in seagrass physiology (e.g., nitrogen uptake, carbon sequestration), their plant and canopy structure (e.g., leaf morphometrics, shoot density, biomass), the size of seagrass meadows (e.g., decrease in cover, disappearance), as well as changes in their associated flora (e.g., epiphyte loads, algal growth) and fauna (e.g., abundance, composition, diversity) (Table 1) and sediment composition and chemistry (e.g., % organic matter, sulphides, primary production; (*Holmer et al., 2008*)). The distance at which effects from a fish farm can be measured vary with the environmental variable being measured and the intensity and duration of farming, husbandry practises and local oceanographic conditions (*Holmer et al., 2008*; *Karakassis et al., 2013*).

*Zostera marina*, commonly known as eelgrass, is the dominant species of seagrass in Atlantic Canada (*Short et al., 2011*). It has been designated an Ecologically Significant Species (ESS) due to its crucial role in providing essential habitat for numerous species and other key ecological services, including nutrient cycling, carbon sequestration, and reduction of wave action (*DFO, 2009a*; *DFO, 2012*). With their extensive root and rhizome system, eelgrass beds also stabilize sediments and provide a rich food source and habitat for infauna communities (*Orth, Heck Jr & Van Montfrans, 1984*; *Boström & Bonsdorff, 1997*). Unlike seagrasses in the Mediterranean Sea that occur at depths down to 40–45 m (*Telesca et al., 2015*), eelgrass meadows along the Atlantic Coast of Canada are found at shallower depths (1–12 m; *DFO, 2009a*) due to stronger light limitation in temperate waters (*Hemminga & Duarte, 2000*). Since open netpen fish farms in Atlantic Canada typically require a minimum depth of 12 m (net depth plus 3 m clearance from bottom; (*Hargrave, 2002*), eelgrass meadows are generally not located directly under, but adjacent to, fish pens.

The potential impacts of finfish aquaculture on eelgrass beds in Canada have been acknowledged for more than a decade (*Vandermeulen, 2005*). Several zones of potential impacts have been identified: eelgrass closest to a finfish farm (Zone A) is likely to be dead or dying and covered in epiphytes, bacterial or fungal mats; eelgrass at some undetermined intermediate distance from a farm (Zone B) is likely to have reduced growth and heavy to moderate epiphyte loads and benthic algal mats; and eelgrass even further from a fish farm (Zone C) is expected to have minimal or even positive growth due to nutrient enrichment (*Vandermeulen, 2005*). To date, however, there have been no quantitative studies in Atlantic Canada to describe the potential impacts of finfish aquaculture on eelgrass habitats, to

**Table 1 Overview of published responses of seagrass (*Posidonia oceanica*) variables (Metrics) at different levels of organization (Level) in the proximity to netpen fish farms in the Mediterranean Sea.**

| Level | Metric | Response | References |
|---|---|---|---|
| Physiological: tissue variables | N and C content in L, R or Rh | increase | *Pérez et al. (2008), Rountos, Peterson & Karakassis (2012)* |
| | P content in L, R or Rh | increase | *Apostolaki et al. (2009)* |
| | S content in R or Rh | increase | *Frederiksen et al. (2007)* |
| | $\delta^{15}$N in L, R or Rh | increase | *Vizzini & Mazzola (2004), Dolenec et al. (2006), Ruiz, Marco-Méndez & Sánchez-Lizaso (2010)* |
| | $\delta^{13}$C in L, R or Rh | increase | *Vizzini & Mazzola (2004), Holmer et al. (2004)* |
| | $\delta^{34}$S in R or Rh | increase | *Frederiksen et al. (2007)* |
| | Sucrose (total non-structural carbohydrates) in R or Rh | decrease | *Delgado et al. (1999), Ruiz, Pérez & Romero (2001), Pérez et al. (2008)* |
| Individual: plant growth | Leaf morphometrics (length or width) | decrease | *Delgado et al. (1999), Apostolaki et al. (2009), Rountos, Peterson & Karakassis (2012)* |
| | Rhizome growth | decrease | *Cancemi, De Falco & Pergent (2003), Marbá et al. (2006)* |
| Population: canopy structure | Shoot density | decrease | *Delgado et al. (1999), Cancemi, De Falco & Pergent (2003), Pergent-Martini et al. (2006), Díaz-Almela et al. (2008), Holmer et al. (2008), Apostolaki et al. (2009), Rountos, Peterson & Karakassis (2012)* |
| | %Cover | decrease | *Ruiz, Pérez & Romero (2001), Cancemi, De Falco & Pergent (2003), Holmer et al. (2008)* |
| | Shoot mortality | increase | *Delgado et al. (1999), Díaz-Almela et al. (2008), Holmer et al. (2008)* |
| | Total Biomass | decrease | *Delgado et al. (1999), Rountos, Peterson & Karakassis (2012)* |
| | AG and BG biomass | decrease | *Delgado et al. (1999), Apostolaki et al. (2009)* |
| | Ratio AG to BG biomass | decrease | *Apostolaki et al. (2009)* |
| Community: associated flora and fauna | Epiphyte load | increase | *Pergent et al. (1999), Cancemi, De Falco & Pergent (2003), Ruiz, Pérez & Romero (2001), Vizzini & Mazzola (2004), Pérez et al. (2008), Rountos, Peterson & Karakassis (2012)* |
| | Epiphyte N and P content and $\delta^{15}$15N | increase | *Pérez et al. (2008), Ruiz, Marco-Méndez & Sánchez-Lizaso (2010), Apostolaki et al. (2007), Apostalaki, Vizzini & Karakassis (2012)* |
| | Microphytobenthos (Chl-*a*) | increase | *La Rosa et al. (2001)* |
| | Phytoplankton (Chl-*a*) | increase | *Pitta et al. (2006), Dalsgaard & Krause-Jensen (2006)* |
| | Annual macroalgal growth | increase | *Dalsgaard & Krause-Jensen (2006)* |
| | Meiofaunal abundance | increase | *La Rosa et al. (2001), Holmer et al. (2008)* |
| | Meiofaunal taxon richness | decrease | *Holmer et al. (2008)* |
| | Macrofaunal abundance | increase | *Terlizzi et al. (2010)* |
| | Macrofaunal diversity ($J'$, $1-\lambda'$) | decrease | *Terlizzi et al. (2010)* |
| | Herbivory | increase | *Pergent et al. (1999), Ruiz, Pérez & Romero (2001), Holmer et al. (2008)* |

**Notes.**
Abbreviations refer to: C, carbon; N, nitrogen; P, phosphorus; S, sulphur; $\delta$, stable isotopes ratios; L, leaves; R, roots; Rh, rhizomes; AG, aboveground; BG, below-ground; Chl-*a*, chlorophyll; J', evenness index; $1-\lambda'$, Simpson's diversity index.

delineate the width of different impact zones, or to assess potential monitoring variables that would aid in the development of specific conservation and management objectives.

As a first step towards addressing these gaps, the objectives of this study were to (i) assess the changes in eelgrass bed structure and associated macroinfaunal communities at

increasing distances from a finfish farm in Port Mouton Bay, and a reference site in adjacent Port Joli Bay, Nova Scotia, (ii) link the observed changes in macroinfauna to changes in environmental and eelgrass bed variables, and (iii) suggest possible monitoring metrics for assessing and managing the impacts of finfish aquaculture on eelgrass habitat. We also place our results in a broader context by comparing them to published eelgrass bed surveys from other sites without fish farms on the Atlantic Coast of Canada (*Cullain et al., 2017*) and to published studies on fish farm effects on seagrass beds in the Mediterranean (Table 1). The Port Mouton fish farm has been in operation since 1995; however, a super chill event killed most fish five months prior to our sampling in 2015, which potentially reduced the immediate short-term but not the long-term impacts being observed. Overall, our study provides insights into changes in eelgrass habitats near a finfish farm in Atlantic Canada and we suggest possible metrics for assessing and monitoring local and broader-scale impacts of fish farms on these ecosystems.

## MATERIALS AND METHODS

### Study area

Our study sites were located along the Atlantic Coast of Nova Scotia (Table 2, Fig. 1). Port Mouton Bay, site of the finfish farm, is a partially sheltered bay covering an area of 55.6 km$^2$ (Fig. 1). Tides, averaging 1.5 m, are semi-diurnal and water depth throughout the bay ranges from 8–18 m. Tidal currents tend to be low (2–3 cm s$^{-1}$, *Gregory et al., 1993*) and surface currents are strongly influenced by winds (*DFO, 2007*; *DFO, 2009b*). During our sampling period (July 14–21, 2015), sea surface temperatures were quite similar at our four sites (12−15 °C) and sampling depth in eelgrass beds ranged from 1.7–2.9 m. These are typical conditions for eelgrass habitats in Nova Scotia, which usually occur at depths of 1–5 m (*Schmidt et al., 2011*; *Cullain et al., 2017*) with optimal growing temperatures ranging from 10−25 °C (*DFO, 2009a*). In general, surficial sediments in Port Mouton and Port Joli Bay are a mix of muddy, sandy and gravelly sand, but in sheltered areas muddy sands predominate (Table 2; *Piper et al., 1986*).

A finfish farm has been operating in Port Mouton Bay since 1995. The current fish farm lease (43°54′54.11″N; −64°48′43.62″W) near Spectacle Island (Fig. 1) occupies an area of 8 hectares (ha) and the sea cages occupy ∼0.58 ha of the lease area (*McIver et al., 2018*). The fish farm was initially stocked with rainbow trout (*Oncorhynchus mykiss*) until ∼2000, followed by Atlantic salmon (*Salmo salar*) until 2009, fallowed from 2010–2012, and stocked with rainbow trout in 2012–2014. The farm lease has been re-licensed for Atlantic salmon and rainbow trout from March 2015 to March 2020 (*NSDFA, 2017*). Information on production at the farm site is deemed proprietary, however it has been estimated at 760 mt annually for 2012–2014, with an estimated 30 mt of dissolved inorganic nitrogen being released from the farm every year (*McIver et al., 2018*). Environmental monitoring at the fish farm indicated that in 2014, one year prior to our sampling, mean sediment sulphides, an indicator or organic enrichment, were 4,000 µM (*NSDFA, 2014*). Sediment enriched with organic waste from fish farms and having sulphide values >3,000 uM are classified as hypoxic indicating redox values <−100 mV and poor macrofaunal diversity (*Hargrave,*

**Table 2  Site names with their abbreviations and associated distance from the finfish farm, bottom temperature, depth and bottom type.** Included are the four study sites sampled in July 2015 (SI, CB, OW, PJ), and 7 other Nova Scotia (NS) sites without finfish farms for comparison (GB, ST, FG, CR, SM, FP, TH) located along the Atlantic Coast of Canada (Fig. 1). The three sites in Port Mouton Bay (SI, CB, OW) were located at varying distances from a finfish farm, and our reference site (PJ) was in adjacent Port Joli Bay. Bottom types include sand (S), mud (M), muddy sand (MS), cobble and sand (CS), and sand and boulders (SB).—means that the distance is not relevant.

| Site (Abbreviation) | Distance (km) | Temp. (° C) | Depth (m) | Bottom type |
|---|---|---|---|---|
| Spectacle Island (SI) | 0.3 | 15 | 2.0 | MS |
| Carters Beach (CB) | 0.7 | 12 | 2.5 | S |
| Old Wharf (OW) | 3 | 14 | 1.7 | MS |
| Port Joli (PJ) | >10 | 15 | 2.9 | SB |
| Green Bay (GB) | – | 14 | 1.4 | M |
| Strawberry Island (ST) | – | 14 | 4.4 | CS |
| Franks George Island (FG) | – | 15 | 4.3 | S |
| Croucher Island (CR) | – | 15 | 3.6 | S |
| Inner Sambro Island (SM) | – | 12 | 4.8 | S |
| False Passage (FP) | – | 12 | 4.6 | S |
| Taylor Head Provincial Park (TH) | – | 10 | 4.9 | SB |

*2010*). In general, Port Mouton Bay is ice-free in the winter, but ice conditions do occur with the most recent event in 2015, five months prior to our sampling. This super chill event killed almost all the fish and the farm has not been restocked. Therefore, there were no fish in the farm cages when we collected our data in July 2015, which may have reduced the potential for immediate short-term impacts being observed, but not the long-term impacts.

### Sampling design and data collection

In accordance with the published literature on the impacts of fish farms on Mediterranean seagrass beds (Table 1), the aims of our study were to test for differences in (1) environmental variables, including sediment organic content, microphytobenthos chlorophyll-*a* concentration, percent cover of epiphytic and benthic annual algae and epiphytic fauna, (2) eelgrass tissue variables, including % nitrogen content and stable isotope ratios of $\delta^{15}$N and $\delta^{13}$C, (3) eelgrass plant and canopy structure, including canopy height, shoot density, percent cover and biomass, and (4) macroinfauna abundance, biomass, species richness and community composition between study sites with increasing distance from the finfish farm in Port Mouton Bay and a reference site in adjacent Port Joli (PJ) Bay (Fig. 1, Table 2). We also compared our result to published data from seven other Nova Scotia sites (NS) without fish farms along the Atlantic Coast (Fig. 1, Table 2; *Cullain et al., 2017*) to check whether our reference site PJ is representative of other NS sites. Finally, we also aimed to (5) link the observed differences in macroinfaunal communities to the environmental, eelgrass tissue and canopy variables using multivariate distance matrices.

Initially, five sampling sites were identified in Port Mouton Bay (Fig. 1). In 2014, the Jackie's Island and Port Mouton Island sites (Fig. 1, historical eelgrass beds) had healthy
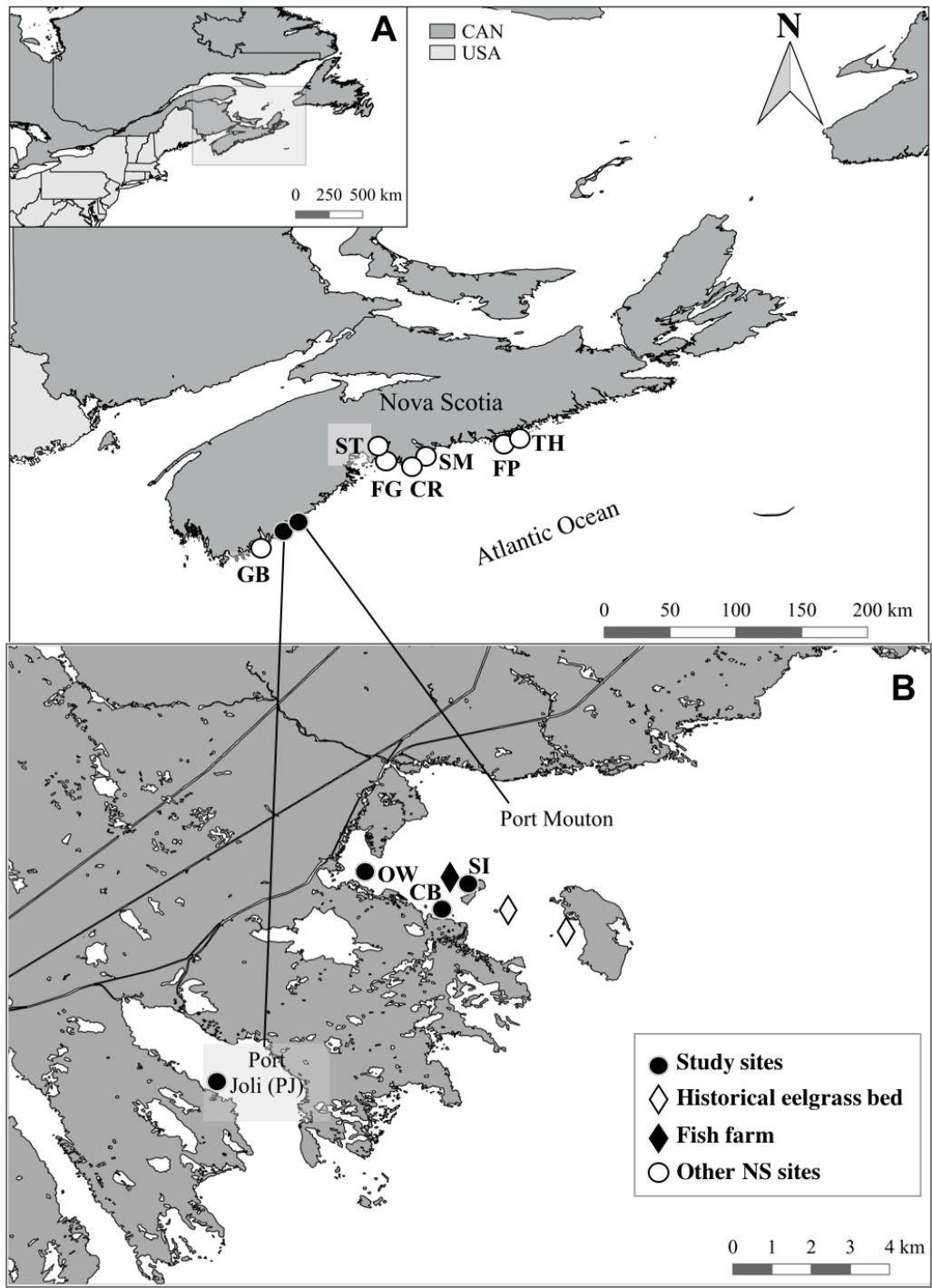

**Figure 1  Location of the study sites and other Nova Scotia (NS) sites for comparison.** All sites located along the Atlantic Coast of Canada (A) with a detailed map (B) of the location of the three study sites and the finfish farm in Port Mouton Bay and the reference site in Port Joli Bay. Historical eelgrass sites indicate areas where eelgrass was no longer present in July 2015. Refer to Table 2 for full site names and details.

eelgrass beds (*Lee, 2014*) that were still visible during our pilot survey (April 2015); however, by July 2015, these sites were highly degraded and had largely disappeared and therefore rejected as sampling sites. The three remaining sites were at varying distances from the finfish farm (Fig. 1, Table 2): an eelgrass bed closest (∼300 m) to the fish farm at Spectacle Island (SI), an eelgrass bed ∼700 m from the fish farm and close to Carters Beach (CB), and an eelgrass bed near an Old Wharf (OW) ∼3,000 m from the fish farm. A reference site (>10 km from the farm) was selected in adjacent Port Joli Bay (PJ) with similar physical and biogeographical conditions (Fig. 1, Table 2). Port Joli Bay is bounded by protected land (Kejimkujik National Park and Thomas Raddall Provincial Park) which restricts human development and nutrient loading in the area (*Nagel et al., 2018*). The seven other NS sites were spread on the southern and eastern shore of Nova Scotia (Fig. 1, Table 2), were generally comparable to the characteristics at the four study sites described above, and had been surveyed in 2013 with the same methodology and timing (mid July to early August) as our study (*Cullain et al., 2017*), except for three sites (SM, ST and CR) where some environmental (microphytobenthos, sediment organic content) and eelgrass tissue variables (%N, $\delta^{13}$C, $\delta^{15}$N) had not been collected. These seven other NS sites had no fish farm or other major point source of pollution close the respective sampling sites (*Cullain et al., 2017*).

All sites were in relatively sheltered, shallow, soft-sediment areas (Fig. 1, Table 2) with eelgrass as the dominant macrophyte (continuous beds > 50 m). Field surveys were conducted in Port Mouton Bay and Port Joli Bay from July 14–21, 2015. At each site, two 50 × 4 m transects were laid approximately 6 m apart parallel to the shore inside the eelgrass bed ≥10 m from the vegetation-bare substrate interface. All sampling was conducted during high tide at the same six locations; 0, 30, and 50 m along the shoreward transect and 5, 25, and 45 m along the seaward transect. Using SCUBA, eelgrass canopy structure (shoot density, canopy height, percent cover) and the percent cover of all epiphytic and benthic macroalgae and epiphytic fauna were assessed using quadrat sampling (0.5 × 0.5 m). A 0.25 × 0.25 m inset was used to count the number of shoots and measure canopy height in the centre of the quadrat by holding the zero end of a measuring tape against the substrate and extending it to the average height of the 80% of the plants (*Duarte & Kirkman, 2001*). The percent cover measures were estimated to the nearest 2% and we considered both sides of all eelgrass blades in the quadrat as habitable space for epiphytic cover. Therefore, if both sides of all the blades in the quadrat were covered with epiphytes, this would represent 100% cover. For the macroalgae, we separated all species used as common indicators of eutrophication for further analysis, including all annual green and brown algae (*Ulva intestinalis*, *Spongomorpha* sp., *Ectocarpus siliculosus*, *Pilayella littoralis*, *Sphaerotrichia divaricata*, *Worm & Lotze, 2006*; *Schmidt et al., 2012*). If species needed further identification, they were bagged and examined under a field dissecting microscope. Similarly, from the epiphytic fauna, we separated species identified as potential indicators of environmental disturbance such as increased turbidity and organic loading (*Ben Brahim et al., 2014*) which in our surveys were primarily hydroids (*Campanularia* sp.). Temperature and depth were recorded on SCUBA dive computers.

Seagrass biomass as well as the abundance of sediment macroinfauna were collected using sediment cores (0.2 m diameter; 0.2 m deep) at each of the six sampling locations along the transects at each site. First, we examined the core to check whether we could observe a redox potential discontinuity (RPD) layer, any black or anoxic sediment and sulfur smell. Next, all aboveground (AG) and belowground (BG) eelgrass tissue was removed, rinsed in a 500 μm sieve to capture any fauna, bagged and kept on ice. Sediment type in each core was recorded (e.g., sand, mud), and the presence of any sulfur smell indicating hypoxia or anoxia. The remainder of the core contents were sieved (500 μm) on site and all macroinfauna species were identified to the lowest possible taxon. If organisms needed further identification, they were kept on ice and brought back to the laboratory and examined under a dissecting microscope. Individuals of each species were counted (abundance m$^{-2}$) and wet weighed (biomass WW g m$^{-2}$). In the laboratory, the eelgrass blades (AG) and roots and rhizomes (BG) were rinsed again and all epiphytes were carefully scraped off the blades. AG and BG tissue was then weighed (WW, g m$^{-2}$) prior to drying in an oven at 60 °C for 48 h and weighed again for dry weight (DW g m$^{-2}$). Once dry, the AG and BG tissues were homogenized separately and a well-mixed 50 mg subsample of each were sent to the University of California Davis Stable Isotope facility to analyze the tissue nitrogen (%N) content as well as the carbon ($\delta^{13}$C) and nitrogen ($\delta^{15}$N) stable isotopes.

Microphytobenthos (MPB) and sediment organic content (SOC) were collected using 60 ml syringe cores (2.6 cm diameter; 2 cm and 5 cm depth, respectively). At each of the six sampling locations, two samples were collected for SOC (volume of sample ∼8.83 mL) and three for MPB (volume of sample ∼3.53 mL). Both SOC samples were placed in a plastic bag and frozen until processed whereas each set of three MPB samples were combined on site, placed in plastic cryovials and stored in liquid nitrogen while in the field and in a freezer (−20 °C) until analysis in the laboratory. In the laboratory, SOC samples were processed according to *Luczak, Janquin & Kupka (1997)*. All MPB samples were processed in a darkened room. Frozen sediment samples were placed in labeled glass scintillation vials with 10 mL of 90% acetone, vortexed for 1 min and then placed back in the freezer to be digested for 24 h. The following day, samples were vortexed for one minute, placed in falcon tubes and centrifuged for 30 min at 3,250 rpm (T. Whitsit, Dalhousie University, pers. comm). The supernatant was subsequently pipetted into clean scintillation vials and measured in a Turner Designs 10005R fluorometer to determine chlorophyll-*a* concentrations.

## Data analysis

Samples from the six cores at all sites were grouped according to their site designation (SI, OW, CB, PJ) except for the seven other sites that were grouped into NS as their site designation and included in the statistical analyses. As such, we had 1 fixed factor (site) with five levels (SI, OW, CB, PJ and NS). We used multivariate permutational analysis of variance (PERMANOVA) to assess the effect of site on the Euclidean distance matrix of each non-independent pair of canopy and environmental variables. If statistically significant ($p \leq 0.05$) differences were found in the paired variables, univariate PERMANOVA was used to assess the effect of site on each of the individual environmental and eelgrass

parameters followed by post-hoc pairwise tests between sites if there was a significant main effect. We also considered significance levels of $p \leq 0.1$ if differences were ecologically relevant with regard to observed effects in the published literature (Table 1; *EFSA Scientific Committee, 2011*). Similarly, univariate PERMANOVA and pairwise tests were used for the independent environmental variables (SOC, MPB) and arcsine transformed zero-adjusted Bray–Curtis similarity matrix for each of annual algae, hydroids and eelgrass % cover. Zero-adjusting dampens the fluctuations of the metric for near-blank samples in an analogous way to the addition of a constant to the log transformation (*Clarke & Gorley, 2015*).

To determine differences in macroinfauna community composition between sites, multivariate PERMANOVA was used on the zero adjusted Bray–Curtis similarity matrix of abundance (density) and biomass separately both with and without the inclusion of the NS sites. Abundance and biomass data were square-root transformed in order to down-weight the influence of highly abundant or large species (*Clarke & Gorley, 2015*). If a significant effect of site was detected, we used post-hoc pairwise tests to determine which sites were significantly different from each other and group average cluster analysis using the centroids of each site to visualize the community data. Univariate PERMANOVA on the Euclidean distance matrix of species richness as well as the square-root transformed zero-adjusted Bray–Curtis similarity matrix of each of total macroinfauna abundance and biomass was used to identify differences in individual summary measures between sites.

To focus in on the potential effects of the fish farm, we excluded the NS sites when using abundance-biomass comparison (ABC) curves, SIMPER and BIOENV analyses. We used ABC curves of the log species rank ($x$-axis) and the cumulative percent dominance ($y$-axis) to compare the $k$- dominance curves for macroinfauna abundance and biomass at each site (*Warwick, 1986*). In unpolluted sites, the biomass curve will be above the abundance curve, in moderately polluted areas the two curves will closely coincide, and in grossly polluted sites the abundance curve will be above the biomass curve (*Warwick, 1986*; *Warwick & Pearson, 1987*). This method expands on the theory by *Pearson & Rosenberg (1978)* where unpolluted sites will have less but larger individuals, but will shift to higher abundances of small opportunistic species as pollution level increases.

To determine which species contributed most consistently (>10%) to the differences in the infauna assemblages between sites near the fish farm, we used similarity percentages (SIMPER) analysis (*Anderson, Gorley & Clarke, 2008*). BIOENV was used to link the overall response of the macroinfauna community to different environmental and eelgrass variables. Prior to these analyses, we examined the correlations among all environmental (SOC, MPB, annual algae, hydroids), eelgrass tissue (AG and BG tissue %N, AG and BG $\delta^{13}$C and $\delta^{15}$N) and eelgrass canopy variables (shoot density, canopy height, percent cover, AG and BG biomass) and those with high correlation ($\geq 0.7$) were never included in the same analysis. If variables were equally correlated (e.g., AG and BG biomass), we chose to include the variable most relevant for infauna (e.g., BG biomass). Therefore, four environmental (SOC, MPB, annual algae, hydroids), three eelgrass tissue (BG %N, BG $\delta^{13}$C, BG $\delta^{15}$N), and four eelgrass canopy (shoot density, canopy height, percent cover, BG eelgrass biomass) variables were used in the analyses. BIOENV provides a non-parametric index rho (ranging from 0
to 1) that indicates how closely different combinations of environment and eelgrass canopy variables explain the multivariate pattern of the macroinfauna community based on the abundance and biomass data and individual SIMPER species. We then used a permutation test to determine the significance level of the sample statistic (rho).

Finally, to combine all variables, we ran a non-metric MDS overlaid with cluster analysis using a normalized Euclidean distance matrix of all environmental, eelgrass tissue and eelgrass canopy variables, as well as macroinfauna total abundance, total biomass and species richness to explore how the sites in Port Mouton Bay and Port Joli Bay cluster relative to the other sites in Nova Scotia. All PERMANOVA, MDS, cluster, ABC, SIMPER and BIOENV analyses were carried out using PRIMER (version 6.1.11) with PERMANOVA+ (version 1.0.1, PRIMER-E; Plymouth Marine Laboratory, Plymouth, UK).

# RESULTS

## Environmental, eelgrass tissue and eelgrass canopy variables

Bottom temperature ranged from $12-15$ °C between the four sites and sampling depth ranged from 1.7–2.9 m (Table 2). Sediment organic content differed between sites (Table 3), with SI, OW and NS having statistically significantly higher organic content than CB and PJ (Fig. 2A). Microphytobenthos did not differ significantly between sites (Table 3), although higher mean values were observed at the three Port Mouton Bay sites compared to PJ and NS (Fig. 2B). There were significant site effects on the cover of annual algae and epiphytic hydroids (Table 3), with CB having the highest cover of annual algae and SI the highest cover of hydroids (Fig. 2C). These observed patterns in environmental variables are in line with those reported in the literature (Table 1).

For above- (AG) and belowground (BG) tissue nitrogen (%N), no statistically significant ($p > 0.05$) but ecologically relevant differences were found (Table 3), with the three Port Mouton sites having higher mean AG %N compared to PJ and NS sites (Fig. 3A). Both AG and BG carbon ($\delta^{13}$C) and nitrogen ($\delta^{15}$N) stable-isotope ratios significantly differed between sites (Table 3), with CB having higher AG $\delta^{13}$C and CB and SI having higher BG $\delta^{13}$C than the other sites (Fig. 3B). For both AG and BG $\delta^{15}$N, SI and OW had lower values than CB, PJ and NS (Fig. 3C). The observed patterns in eelgrass tissue variables except $\delta^{15}$N are in line with the published literature (Table 1).

Multivariate PERMANOVA detected no statistically significant differences for eelgrass shoot density and canopy height across sites (Table 3; Figs. 4A, 4B), but there was a tendency for lower shoot density at the sites closer to the fish farm (SI and CB) than the other sites including NS (Fig. 4A). Eelgrass cover and both AG and BG biomass decreased at sites closer to the fish farm (Figs. 4C, 4D), which was significant for eelgrass cover (Table 3). Despite the significant multivariate site effect on AG and BG biomass, the univariate tests were only $p < 0.1$ and showed that SI and CB had lower AG and BG biomass than PJ and NS. These observed patterns on eelgrass canopy variables are in line with the published literature (Table 1).

**Table 3  Results from multivariate and univariate PERMANOVAs on the effect of site on environmental variables, eelgrass tissue and eelgrass canopy structure across sites.** Included are the four study sites and an average of the other NS sites for comparison along the Atlantic Coast of Canada. Multivariate analyses were only performed for non-independent variables, and only followed up with univariate analyses if statistically significant differences ($p \leq 0.05$, bolded) were found or ecologically relevant patterns from other studies were observed in our data (e.g., %$N$).

| | DF | RDF | Multivariate pseudo-F | Multivariate p-value | Univariate pseudo-F | Univariate p-value |
|---|---|---|---|---|---|---|
| **Environmental variables:** | | | | | | |
| Sediment organic content | 4 | 41 | – | – | 3.18 | **0.016** |
| Microphytobenthos | 4 | 41 | – | – | 0.55 | 0.72 |
| Annual algae cover | 4 | 61 | 10.9 | **0.001** | 10.9 | **0.001** |
| Hydroid cover | | | | | 8.48 | **0.001** |
| **Tissue variables:** | | | | | | |
| Tissue % $N$ - Above | 4 | 41 | 1.61 | 0.12 | 2.75 | **0.05** |
| - Below | | | | | 0.67 | 0.61 |
| Tissue $\delta^{13}$C - Above | 3 | 41 | 7.00 | **0.001** | 11.7 | **0.001** |
| - Below | | | | | 3.96 | **0.011** |
| Tissue $\delta^{15}$N - Above | 3 | 17 | 7.38 | **0.001** | 6.68 | **0.001** |
| - Below | | | | | 8.14 | **0.001** |
| **Eelgrass bed structure:** | | | | | | |
| Shoot density | 4 | 61 | 1.44 | 0.20 | – | – |
| Canopy height | | | | | – | – |
| Percent cover | 4 | 61 | – | – | 2.81 | **0.037** |
| Biomass - Above | 4 | 61 | 2.28 | **0.02** | 2.46 | 0.063 |
| - Below | | | | | 2.11 | 0.083 |

## Macroinfauna community

A total of 36 macroinfauna genera were identified across all sites, 26 of which were identified to the species level. Univariate PERMANOVA detected a difference in the summary measures of total abundance (pseudo-$F_{4,60} = 2.15$, $p = 0.056$) and total biomass (pseudo-$F_{4,60} = 2.68$, $p = 0.023$), but not for species richness (pseudo-$F_{4,60} = 0.49$, $p = 0.73$) of macroinfauna between sites (Fig. 5). Both the abundance and biomass of CB were significantly lower than OW and NS but the variability in the other sites precluded significant differences (Figs. 5A, 5B). Overall the three sites in Port Mouton Bay tended to have lower macroinfaunal biomass compared to PJ and NS (Fig. 5B).

The ABC curves for cumulative dominance for the sites in Port Mouton and Port Joli showed that macroinfauna biomass was distinctly above the abundance curve for both CB and PJ (Figs. 6B, 6D), indicating unpolluted conditions. The biomass and abundance curves for SI approached each other but did not overlap (Fig. 6A), suggesting that this site is approaching moderately polluted conditions. The OW site was the only site where the abundance curve lay above the biomass curve (Fig. 6C), indicating polluted conditions.

Macroinfauna community composition based on both abundance and biomass did differ significantly between sites (multivariate PERMANOVA: pseudo-$F_{5,60} = 2.0$, $p = 0.002$ and pseudo-$F_{4,60} = 1.91$, $p = 0.001$; respectively). Cluster analysis of centroids based on infauna

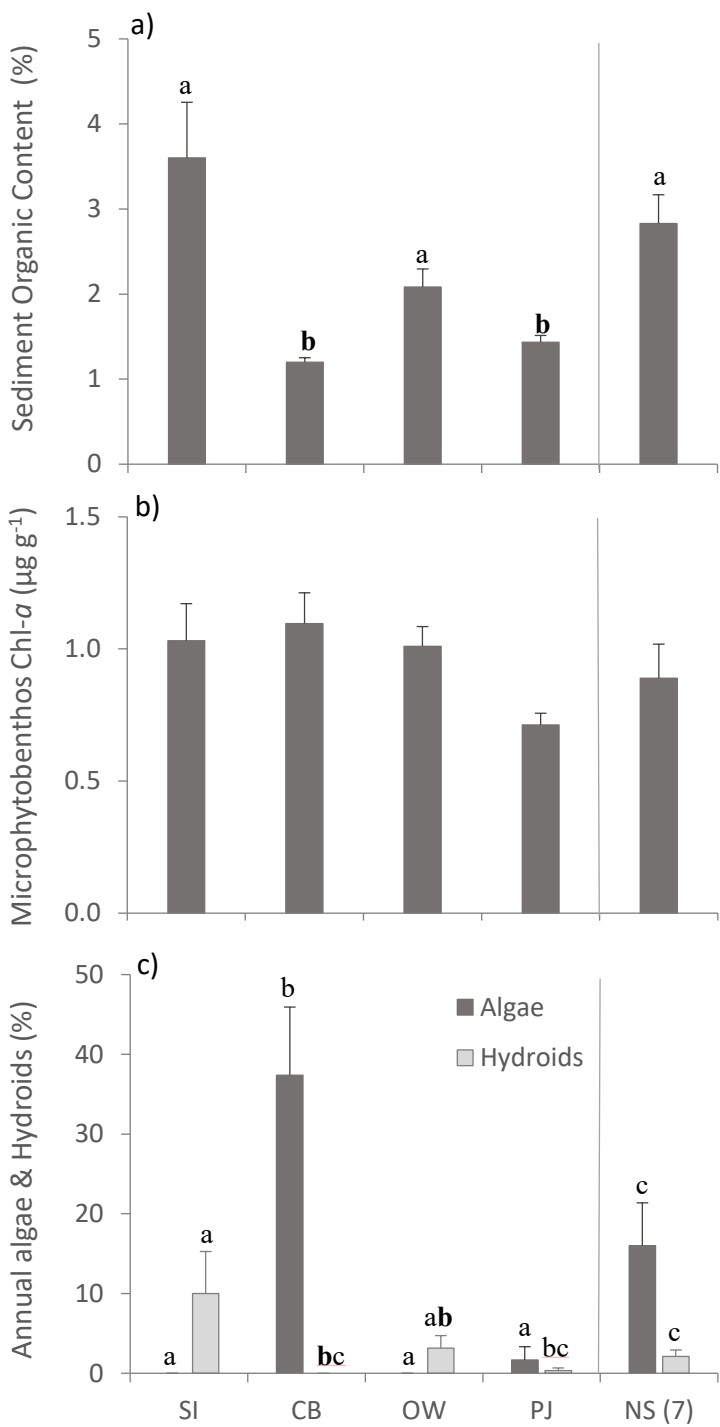

**Figure 2** **Environmental variables across the four study sites.** Environmental variables (mean ± SE) across the four study sites (from left to right: increasing distance from fish farm and PJ reference site) and an average of the other NS sites for comparison: (A) sediment organic content, (B) microphytobenthos chlorophyll - a concentration, and (C) percent cover of epiphytic and benthic annual algae and epiphytic hydroids. Lowercase letters indicate significant differences ($p \leq 0.05$). If the lowercase letters in the same group are bolded, there is a marginally non-significant difference between those sites (i.e., $p$-value between 0.05–0.1) for that variable. Number in brackets beside NS indicates the number of sites included in mean and statistical analyses. Refer to Table 2 for site abbreviations and details.

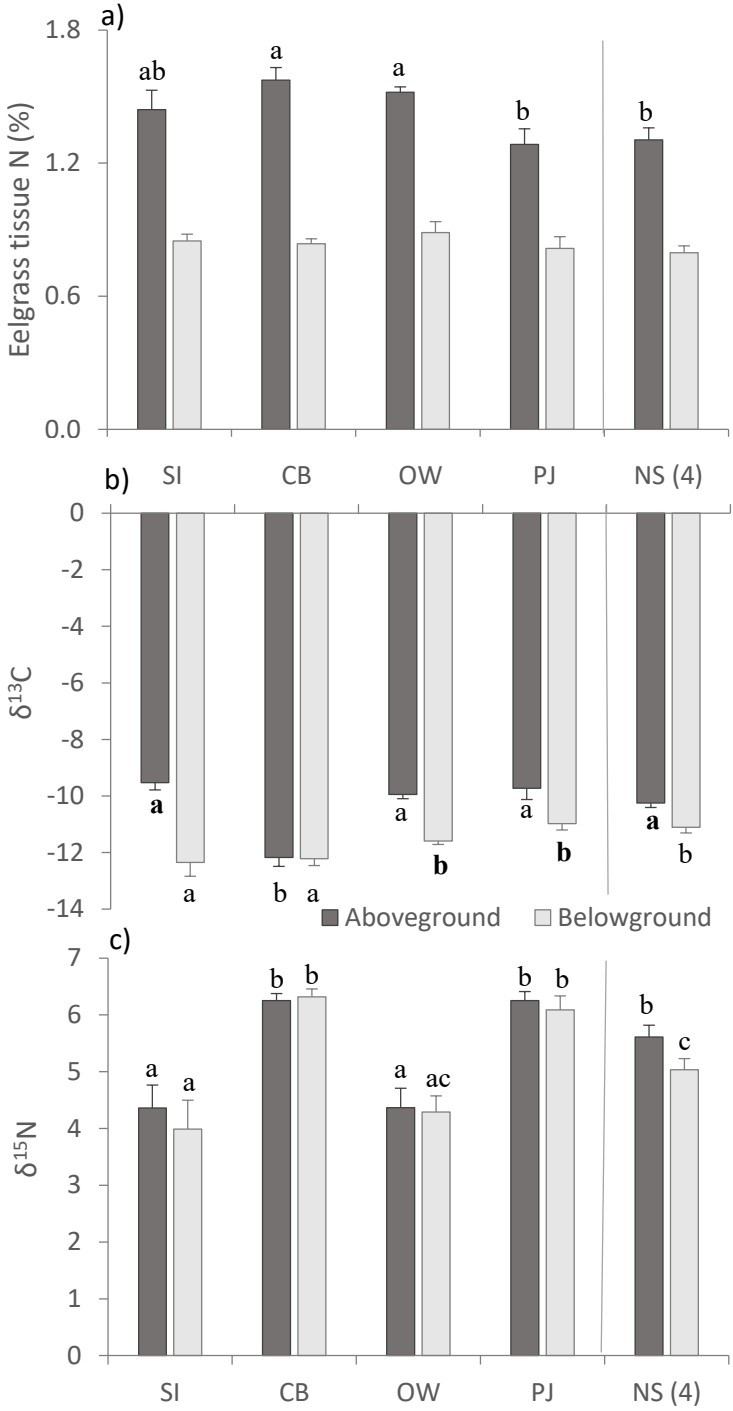

**Figure 3** **Eelgrass tissue variables (mean ± SE) across the four study sites (from left to right: increasing distance from farm and PJ reference site) and an average of the other NS sites for comparison.** (A) eelgrass tissue nitrogen content (%N) and stable isotope ratios of (B) carbon (δ13C) and (C) nitrogen (δ15N) in above—and belowground tissues. Lower case letters indicate significant differences ($p \leq 0.05$). If the lower-case letters in the same group are bolded, there is a marginally non-significant difference between those sites (i.e., $p$-value between 0.05–0.1) for that variable. Number in brackets beside NS indicates the number of sites included in mean and statistical analyses. Refer to Table 2 for site abbreviations and details.

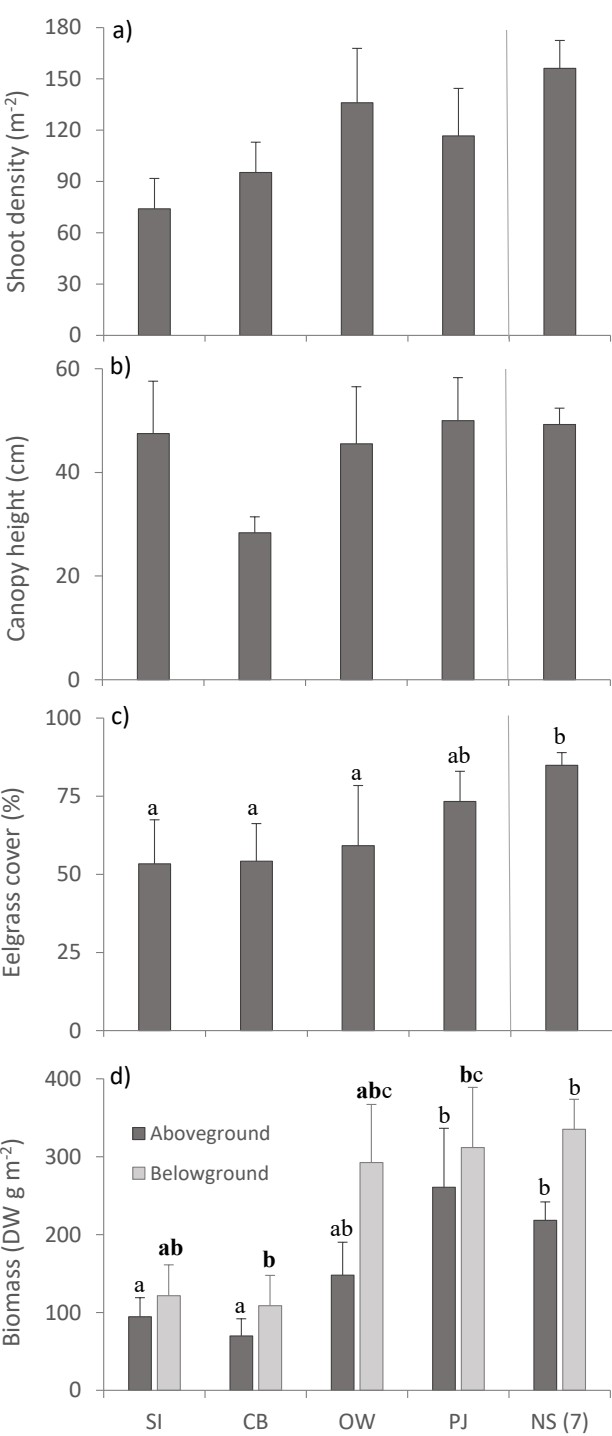

**Figure 4  Eelgrass canopy structure (mean ± SE) across the four study sites.** From left to right: increasing distance from farm and PJ reference site, and an average of the other NS sites for comparison: (A) shoot density, (B) canopy height, (C) percent cover, and (D) above- and belowground biomass. Lower case letters indicate significant differences ($p \leq 0.05$). If the lower-case letters in the same group are bolded, there is a marginally non-significant difference between those sites (i.e., $p$-value between 0.05–0.1) for that variable. Number in brackets beside NS indicates the number of sites included in mean and statistical analyses. Refer to Table 2 for site abbreviations and details.

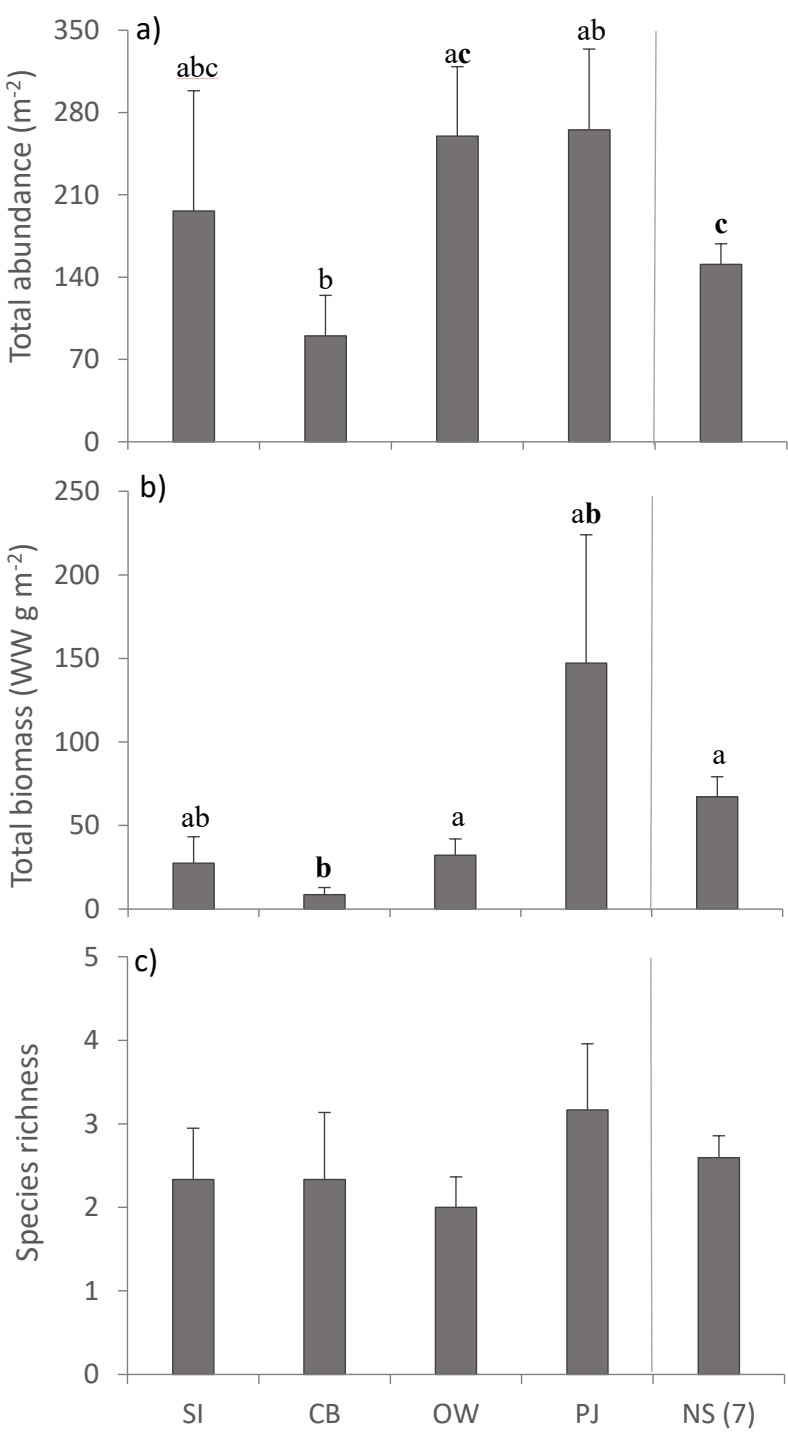

**Figure 5** **Summary measures (mean ± SE) of the macroinfauna community across the four study sites.** From left to right: increasing distance from farm and PJ reference site, and an average of the other Nova Scotia (NS) sites for comparison: (A) total abundance, (B) total biomass, and (C) species richness. Lowercase letters indicate significant differences ($p \leq 0.05$). If the lowercase letters in the same group are bolded, there is a marginally non-significant difference between those sites (i.e., $p$-value between 0.05–0.1) for that variable. Number in brackets beside NS indicates the number of sites included in mean and statistical analyses. Refer to Table 2 for site abbreviations and details.

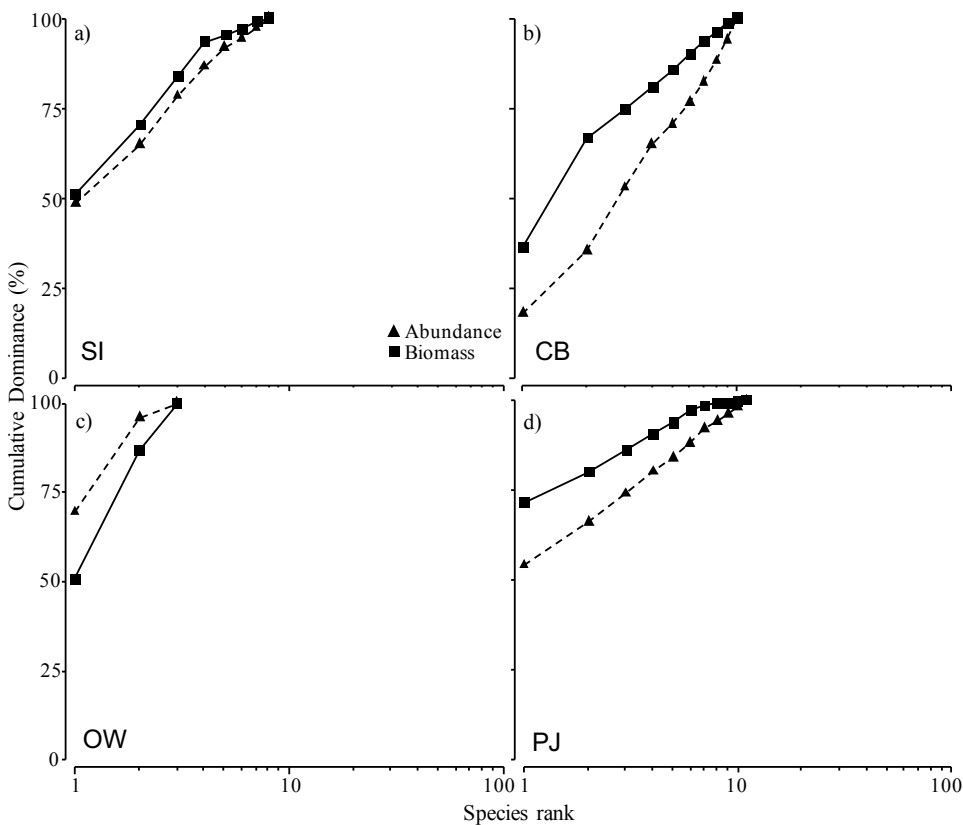

**Figure 6** Abundance-biomass comparison (ABC) curves using cumulative dominance for infauna species for the four study sites at increasing distances from the finfish farm in Port Mouton Bay. (A) Spectacle Island (SI), (B) Carters Beach (CB), (C) Old Wharf (OW), and (D) the reference site Port Joli Bay (PJ) in Nova Scotia, Canada.

abundance data (Fig. 7A) showed a clustering of SI and OW, which were the more polluted sites as identified by the ABC curves, while cluster analysis of infauna biomass data (Fig. 7B) showed a clustering of SI and CB, the two sites closest to the fish farm. In both cases, PJ clustered with NS indicating that it is a representative control site. When we excluded the other NS sites to more closely examine the effects of the fish farm, the PERMANOVA and cluster analyses based on abundance and biomass yielded similar results.

The main species identified by SIMPER contributing >10% of differences in abundance and biomass between sites included three polychaetes: *Clymenella torquata*, *Capitella capitata,* and *Nephtys* sp. with the addition of *Amphitrite* sp. for biomass. However, the contribution of each species to the community differed when considering abundance or biomass, respectively (Table 4). *C. torquata* dominated the community in abundance across all sites except CB, and in biomass except in PJ, where *Amphitrite* sp. was the dominant species (Table 4). The opportunistic polychaete *C. capitata* only occurred at the two more polluted sites in Port Mouton Bay (SI, OW), with the highest abundance and biomass closest to the fish farm (Table 4). The polychaete *Nephtys* sp. also showed higher abundance at the two more polluted sites (SI, OW). In turn, *Amphitrite* sp. as well

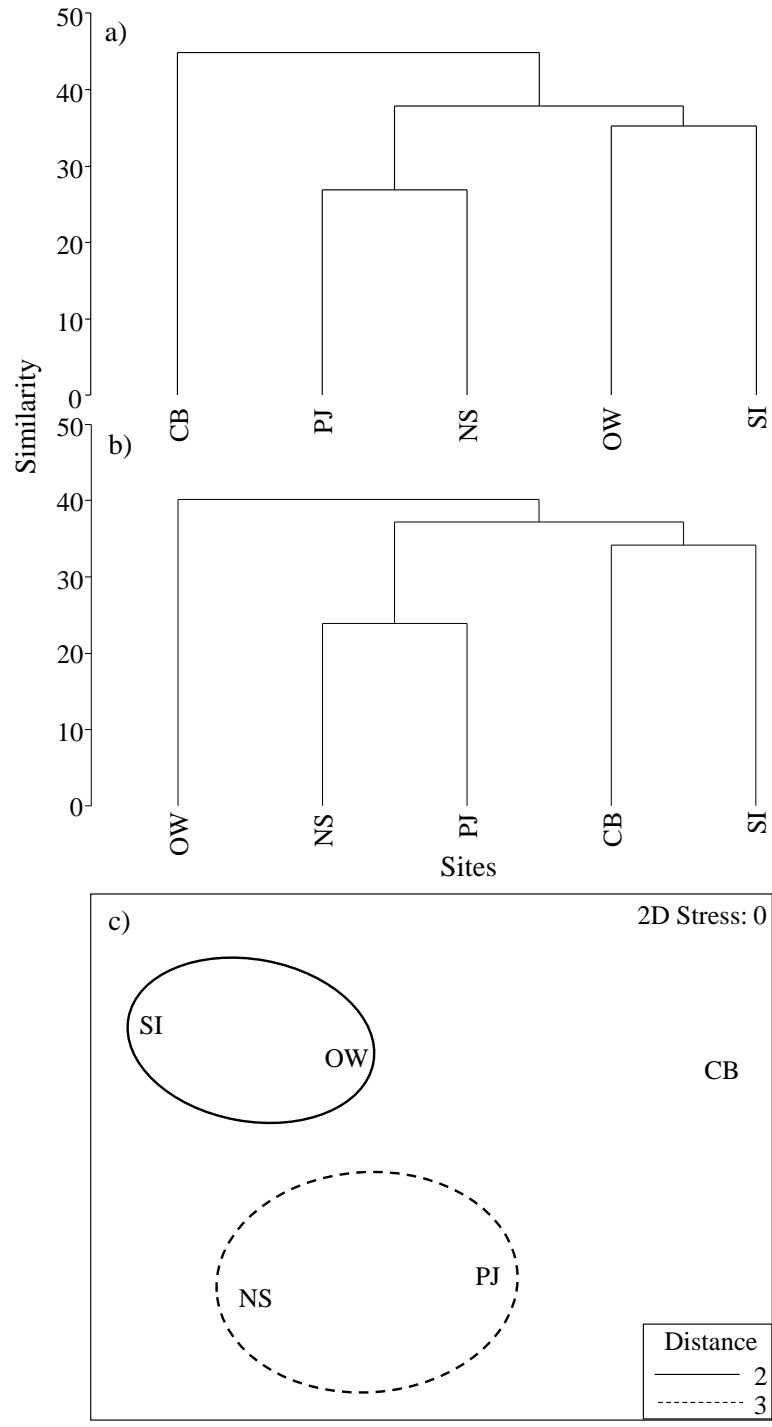

**Figure 7 Cluster analysis using infauna community centroids.** Based on (A) abundance and (B) biomass as well as (C) nMDS with overlaid clusters of all environmental, eelgrass tissue and eelgrass canopy variables as well as total infauna abundance, biomass and richness at the four study sites and an average of the other NS sites for comparison along the Atlantic Coast of Canada. See Table 2 for site abbreviations and details. Distance in (C) refers to the Euclidean distance between centroids.

**Table 4  Mean abundance (ABUN; m⁻²) and biomass (BIOM; g m⁻²) of the most abundant macroinfauna species (±SE) at the four study sites.** From left to right: increasing distance from fish farm and control site (PJ), in Nova Scotia, Canada. Refer to Table 2 for site names and details. Species identified in the SIMPER analysis as contributing to ≥10% of the difference between sites are in bold.

| Species | SI | | CB | | OW | | PJ | |
|---|---|---|---|---|---|---|---|---|
| | ABUN | BIOM | ABUN | BIOM | ABUN | BIOM | ABUN | BIOM |
| *Amphitrite* sp. | 0 | 0 | 0 | 0 | 0 | 0 | 31.9 (21.8) | 105.6 (68.0) |
| *Capitella capitata* | 26.5 (12.8) | 5.3 (2.4) | 0 | 0 | 10.6 (6.7) | 4.3 (3.7) | 0 | 0 |
| *Cerastoderma pinnulatum* | 0 | 0 | 15.9 (7.1) | 0.7 (0.3) | 0 | 0 | 0 | 0 |
| *Clymenella torquata* | 95.5 (83.1) | 14.0 (13.8) | 5.3 (5.3) | 2.7 (2.7) | 180.5 (58.5) | 16.5 (7.6) | 143.3 (39.2) | 12.2 (4.9) |
| *Nephtys* sp. | 31.9 (16.5) | 2.6 (1.3) | 10.6 (6.7) | 3.2 (2.2) | 69.0 (15.2) | 11.6 (3.4) | 15.9 (10.9) | 4.8 (4.2) |
| *Ophelia* sp. | 0 | 0 | 0 | 0 | 0 | 0 | 21.2 (15.8) | 6.4 (5.2) |
| *Tellina agilis* | 5.3 (5.3) | 0.2 (0.2) | 15.9 (10.9) | 0.4 (0.3) | 0 | 0 | 0 | 0 |

as *Ophelia* sp. only occurred at the reference site PJ, whereas the bivalves *Cerastoderma pinnulatum* and *Tellina agilis* were most abundant at CB (Table 4).

### Linking the environment to the macroinfauna community

BIOENV was used to determine any associations between the Euclidean distance of environmental, eelgrass tissue, and eelgrass bed structure variables (SOC, MPB, annual algae, hydroids, BG %N, BG $\delta^{15}$N, BG $\delta^{13}$C, eelgrass shoot density, canopy height, percent cover, and BG biomass) and the Bray–Curtis similarity matrix of macroinfauna community structure (Table 5). The community assemblage based on abundance data was most correlated to BG %N, BG $\delta^{13}$C and the percent cover of annual algae (%A), and the assemblage based on biomass data to BG eelgrass biomass, BG %N and BG $\delta^{13}$C (Table 5). Both the abundance and biomass of *C. capitata* were most correlated to the percent cover of hydroids (%H), while the next best correlations also included SOC and MPB. In contrast, the other species were more correlated with eelgrass bed structure, with BG eelgrass biomass being the most important (Table 5).

Combining all environmental, eelgrass tissue and eelgrass bed structure variables as well as total macroinfauna abundance, biomass and species richness in a nMDS and cluster analysis revealed a distinct cluster of the reference site (PJ) and NS, and another cluster of the more polluted sites, SI and OW (Fig. 7C).

## DISCUSSION

Our study quantitatively assessed the potential impacts of finfish aquaculture on eelgrass bed structure and associated macroinfaunal communities in Atlantic Canada and framed these results in a broader context of fish farm impacts on seagrass beds elsewhere. Our results suggest higher nutrient and organic enrichment, higher algae and epiphyte loads, lower eelgrass cover and biomass, and lower macroinfauna biomass closer to the fish farm. Moreover, the macroinfauna community showed significantly altered species composition. Importantly, the opportunistic polychaete *Capitella capitata,* a known indicator of polluted conditions, specifically organic enrichment (*Pearson & Rosenberg, 1978*) reached its highest abundance close to the fish farm at Spectacle Island (SI), and also occurred at an Old

**Table 5 BIOENV results for the macroinfauna community and species identified by SIMPER analysis.**
Included is abundance data (above the line) and biomass data (below the line) with environmental variables (percent cover of annual algae (%A) and hydroids (%H), sediment organic content (SOC)), eelgrass tissue (belowground (BG) %N, $\delta^{15}$N and $\delta^{13}$C) and eelgrass canopy structure (BG biomass, Shoot density, Canopy height, Eelgrass cover) at the four study sites in Nova Scotia, Canada. Significant ($p \leq 0.05$) correlations are bolded.

|  | Best correlated variable(s) | $\rho$ | *p*-value |
|---|---|---|---|
| Community abundance | BG %N, BG $\delta^{13}$C, %A | 0.45 | **0.02** |
| *Capitella capitata* | %H | 0.45 | **0.04** |
|  | SOC, %H | 0.44 |  |
|  | MPB, %H | 0.44 |  |
| *Clymenella torquata* | BG biomass | 0.47 | **0.01** |
| *Nephtys* sp. | BG biomass, Shoot density, SOC, Eelgrass cover | 0.21 | 0.13 |
| Community biomass | BG biomass, BG %N, BG $\delta^{13}$C | 0.40 | **0.04** |
| *Amphitrite* sp. | BG biomass | 0.21 | 0.82 |
| *Capitella capitata* | %H | 0.50 | **0.02** |
|  | SOC, %H | 0.46 |  |
| *Clymenella torquata* | BG biomass | 0.43 | **0.01** |
| *Nephtys* sp. | BG biomass, Shoot density | 0.17 | 0.38 |

Wharf site (OW) with a history of pollution. This was strongly correlated with higher loads of epiphytic suspension-feeding hydroids and higher sediment organic content pointing to organic enrichment at these sites. The combined assessment of multiple eelgrass tissue, canopy and environmental variables was important in evaluating ecological changes, particularly differences between impacted and unimpacted ecosystems. Overall, the observed patterns at our study sites are in line with published results on the impacts of fish farms on Mediterranean seagrass beds (Table 1). We discuss possible metrics for assessing and monitoring local and broader-scale impacts of nutrient and organic enrichment from fish farms on eelgrass ecosystems.

## Environmental variables

Several physical and biogeochemical factors are known to influence eelgrass growth and survival, including wave exposure, sediment type and water quality (*Frederiksen et al., 2004*; *Vandermeulen, 2005*). While all our study sites were in relatively sheltered areas with similar wave exposure, SI and OW sites were the most sheltered, with SI nestled behind an island and OW positioned in the inner most part of the bay. Current speeds in Port Mouton Bay are generally low (2–3 cm s$^{-1}$, *Gregory et al., 1993*) and flushing times long (114 h, *Nagel et al., 2018*), which could lead to higher sedimentation rates compared to adjacent Port Joli (PJ) Bay, which has faster flushing times (53 h, *Nagel et al., 2018*). We found the highest amount of organic matter in sediments closest to the fish farm at SI, followed by OW, while Carters Beach (CB) inside Port Mouton Bay and the PJ reference site had much lower sediment organic content. In contrast, the other NS sites also had relatively high organic content. Both SI and OW have been affected by long-term anthropogenic activities. The (abandoned) Old Wharf is in an area of higher coastal development with some direct discharge of domestic sewage and a former fish processing plant. The fish farm at SI has

been in operation, on and off, since 1995. Although no redox potential discontinuity (RPD) layer was observed at any of the sites during sampling, strong sulfur smell and dark black sediments were observed while sampling at SI and OW.

A potential early indicator of nutrient enrichment is the increased concentration of microphytobenthos, composed primarily of diatoms and cyanobacteria (*Lever & Valiela, 2005*). Although not statistically significant, we observed higher mean microphytobenthos chlorophyll-*a* concentrations at sites within Port Mouton Bay compared to the reference PJ and other NS sites, in line with expected patterns from the published literature (Table 1). The enhanced microphytobenthos productivity in Port Mouton Bay may be a result of the overall higher nutrient loading from greater human development in the watershed and the finfish farm in the bay (*McIver et al., 2018*) compared to the more protected and less developed watershed in Port Joli Bay with much lower nutrient loading (*Nagel et al., 2018*). Under enhanced nutrient loading, increased microphytobenthos concentration can shift sediment conditions from being autotrophic to net heterotrophic, where sediments become increasingly hypoxic or anoxic, sulphides accumulate, and denitrification and mineralization are enhanced (*Meyer-Reil & Köster, 2000*; *Sundbäck et al., 2004*; *Hardison et al., 2013*). Seagrasses can grow in low sediment oxygen conditions and some species such as *Z. marina* can tolerate higher levels of porewater sulphides (*Hasler-Sheetal & Holmer, 2015*); however, the combination of hypoxia and high sulphides (100–1,000 μM) has been shown to affect *Z. marina* growth and survival, specifically reducing photosynthetic activity, decreasing biomass, and increasing decay in the meristematic region of the plant (*Holmer & Bondgaard, 2001*).

Eelgrass tissue nitrogen content and stable-isotope ratios are commonly used to trace the amount and source of nitrogen, respectively, within seagrass ecosystems (*Hemminga & Duarte, 2000*; *Dolenec et al., 2006*; *Ruiz, Marco-Méndez & Sánchez-Lizaso, 2010*). The higher mean tissue nitrogen content within Port Mouton Bay compared to the reference PJ and other NS sites also point to higher nutrient availability within Port Mouton Bay, possibly due to higher nutrient loading (*Nagel et al., 2018*). However, this seemed to be distributed throughout the bay, as there was no elevated tissue content at the SI site closest to the fish farm. We also did not find elevated stable isotope $\delta^{15}N$ values at the fish farm site. Generally, wastewater from human or animal waste has a higher $\delta^{15}N$ signature (ratio of $^{15}N/^{14}N$) of 8–10‰ up to 20‰ (*Lepoint, Dauby & Gobert, 2004*; *Schubert et al., 2013*), and fish farm waste (feces, mucus, pellets) has $\delta^{15}N$ values of 6.5–10.5‰ (*Dolenec et al., 2006*; *Sarà et al., 2006*; *Ruiz, Marco-Méndez & Sánchez-Lizaso, 2010*). Several studies have found significantly elevated $\delta^{15}N$ at varying distances (250–900 m) up to >1,000 m from fish farms (Table 1; *Holmer et al., 2007*; *Ruiz, Marco-Méndez & Sánchez-Lizaso, 2010*; *García-Sanz et al., 2011*). These values depend on hydrographical characteristics (e.g., depth, current speeds) at and around the farm site, the scale of farm production and feeding efficiencies (*Sarà et al., 2006*; *Holmer et al., 2007*). At our sites, $\delta^{15}N$ values of 4–6.5‰ were within the range of natural variation in seagrass ecosystems (*Hemminga & Mateo, 1996*; *Lepoint, Dauby & Gobert, 2004*) and comparable to values reported in other Atlantic Canadian bays (*McIver, Milewski & Lotze, 2015*). This may be explained by the fact that production at the fish farm ceased 5 months prior to our sampling in July due to

a super chill event that killed all the fish, allowing time for the $\delta^{15}N$ in the eelgrass tissue to be adequately used or cycled within the system. This may also explain why we did not measure enhanced tissue nitrogen content directly at the Spectacle Island site, but rather throughout Port Mouton Bay.

Stable carbon isotope ratios ($\delta^{13}C$) are used for studying the source and fate of organic carbon in ecosystems and food webs. Under increased nutrient and organic loading which, among other effects, increases sediment organic content and decreases light conditions, the balance in carbon utilization in seagrasses shifts from inorganic carbon fixation (photosynthesis) toward belowground consumption of organic carbon (respiration) (*Holmer & Bondgaard, 2001*; *Holmer et al., 2004*). A more negative isotopic signature represents the input of $^{13}C$-depleted carbon from the decomposition of organic material (*Hemminga & Mateo, 1996*). We found less negative $\delta^{13}C$ further away from the finfish farm particularly in belowground roots and rhizomes. Similar patterns of less negative $\delta^{13}C$ further away from the source have been observed in other seagrass beds which receive organic material from fish farms (*Vizzini & Mazzola, 2004*; *Holmer et al., 2004*) or land run-off (*Hemminga & Mateo, 1996*; *Hemminga & Duarte, 2000*). Thus, while the nitrogen signal may have disappeared due to the fish farm not being stocked at the time of sampling, the organic carbon signal was still visible.

Changes in seagrass physiology (e.g., tissue nutrient content and $\delta^{15}N$) in response to changes in nutrient supply or environmental quality (e.g., increased turbidity, decreased light intensity) occurs more quickly (days and weeks) than changes to morphology (e.g., biomass, shoot density) (*Grice, Loneragan & Dennison, 1996*; *Longstaff & Dennison, 1999*). Under high sediment sulphide conditions (commonly associated with organic loading *Pearson & Rosenberg, 1978*), however, both physiological (non-structural carbohydrates and $\delta^{13}C$ isotope signatures) and morphological (dead leaves and rot in the meristematic region) have been observed within 3 weeks (*Holmer & Bondgaard, 2001*). Physiological seagrass measures, therefore, can be good early indicators of change (both deterioration or recovery) and can be used to detect changes to specific stressors (*Roca et al., 2016*).

## Eelgrass bed structure

Frequent responses of seagrasses to increased nutrient and organic loading are decreases in shoot density, biomass and cover, and increases in canopy height (*Short et al., 2011*; *Schmidt et al., 2012*). Our results suggest lower mean shoot density, cover, and biomass closer to the fish farm compared to the reference PJ and other NS sites, similar to other studies on the impacts of fish farm effluent on seagrasses (Table 1; *Pergent-Martini et al., 2006*). A synthesis of the effects of Mediterranean fish farms reported that the most important process affecting *Posidonia oceanica* was the sedimentation of organic material (*Holmer et al., 2008*). Estimated dispersion distances from fish farms are variable, but the furthest distances of organic-enriched material has not exceeded 1,000 m (*Sarà et al., 2006*; *Holmer et al., 2007*). For the Mediterranean, *Díaz-Almela et al. (2008)* proposed a rate of 1.5 g organic matter m$^{-2}$ d$^{-1}$ as a threshold to protect *P. oceanica* from the impacts of fish farms. Although we did not measure sedimentation rates, the development of a comparable threshold value for eelgrass habitat in Atlantic Canada would be a valuable
tool for protecting eelgrass from the impacts of finfish aquaculture as well as other anthropogenic activities.

## Macroinfauna

The ABC curves allowed us to use the total macroinfauna abundance and biomass to examine the sites based on a pollution gradient (*Warwick, 1986*). Our results indicate that OW is considered the most polluted site, followed by SI, while the reference site PJ can be considered unpolluted. The location of OW may explain its polluted status as this site has a decades-long, cumulative history of coastal development including direct sewage outflow and a former fish processing plant which is no longer in operation. The fish farm near SI, on the other hand, has been in operation on and off for 19 (1995–2015) years while the reference site PJ has no industrial development. SI, however, did show the two ABC curves approaching each other, indicating that the site may be transitioning to/from a polluted state (*Warwick, 1986*). Since our study sites were only sampled during one time period, it would be important and valuable from a management perspective to monitor changes in these ABC curves seasonally and over the years.

Summary measures of the macroinfauna community suggested lower total abundance and biomass at CB, and potentially lower biomass at the other Port Mouton Bay sites compared to the reference PJ and other NS sites but with high variability. This is likely due to the significant changes in species composition, with some species increasing and others decreasing (Table 4). Thereby, two polychaete species, *C. capitata* and *C. torquata,* were the main contributors to the differences among sites based on both abundance or biomass. The higher abundance and biomass of *C. capitata* at SI, followed by OW, is not surprising as *C. capitata* has long been associated with sediment organic enrichment from human activities (e.g., sewage, seafood, and wood-processing facilities) including finfish aquaculture (*Mazzola et al., 2000*; *Holmer, Wildish & Hargrave, 2005*; *Martinez-Garcia et al., 2013*).

In their study on the effects of fish farming on soft-bottom polychaete assemblages, *Martinez-Garcia et al. (2013)* reported that tolerance to higher total dissolved sulphides, silt and clay fractions, and sediment stable isotope nitrogen signatures were the main sediment factors that distinguished the occurrence of polychaete families such as Capitellidae compared to families that are more sensitive (meaning a decrease in abundance) to fish farm pollution such as Maldanidae and Nephtyidae, although species in these families were also found in low impact areas. These latter families include two polychaete species, *Clymenella* sp. and *Nephtys* sp. respectively, found in variable abundance at our study sites. *Nephtys* sp. abundance has been found to fluctuate along a stress or pollution gradient in a bimodal pattern (*Pearson, Gray & Johannessen, 1983*). The lowest numbers were found next to pollutant sources, followed by higher numbers ∼0.5 km further out, then a drop in numbers 2 km away and finally an increase and stabilization ∼6 km away (*Pearson, Gray & Johannessen, 1983*). In our study, we did not discern this pattern for *Nephtys* sp. as our sampling design was not on the same spatial or temporal scale as reported in *Pearson, Gray & Johannessen (1983)*; however, we found higher number of *Nephtys* sp. at SI (∼0.3 km) followed by lower numbers at CB (∼0.7 km) (Table 4). *C. torquata* is viewed as an

important macroinfaunal species in *Z. marina* habitat because higher densities of these worms (>192 worms m$^{-2}$) are responsible for adding topographical relief to sediments which in turn is more effective in trapping and burying *Z. marina* seeds and, hence, improving recruitment (*Luckenbach & Orth, 1999*). In our study *C. torquata* abundance was lowest at the sites closest to the fish farm (SI and CB) suggesting a potential recruitment risk to the eelgrass beds in these areas that should warrant further monitoring.

Other species in our data set such as *Amphitrite* sp. and *Ophelia* sp. were present only at our reference site (PJ) and the mollusc *Tellina agilis* was present only at two sites closer to the fish farm (SI and CB). The occurrence and distribution of these species is not readily explained without further study. Apart from *C. capitella,* which is a widely used indicator of organic enrichment, a potential other biological indicator to assess the impact of fish farm waste on macroinfaunal in *Z. marina* habitat could include *C. torquata*.

## Linking the environment/eelgrass structure to infauna community composition

Our results suggest that eelgrass bed structure, specifically belowground biomass, was a good predictor of community and individual species abundance or biomass, but also shoot density and cover. Moreover, tissue nitrogen as an indicator of nutrient enrichment, $\delta^{13}$C as an indicator for enhanced decomposition of organic material, and the cover of epiphytic and benthic annual macroalgae (%A) as a proxy for eutrophication could be linked to community composition based on biomass or abundance. The correlation between macroinfauna composition and belowground eelgrass biomass as well as detritus is well documented and has been linked to the role eelgrass roots and rhizomes have in accumulating and stabilizing sediments which in turn provide habitat and food to the associated detritivore-dominated infauna, releasing oxygen to the sediments and providing protection from predators (*Orth, Heck Jr & Van Montfrans, 1984*; *Lee, Bailey-Brock & McGurr, 2006*; *Boström, Jackson & Simenstad, 2006*). Thereby, sediment grain size and stability can covary with below ground biomass and shoot density to influence macroinfaunal assemblages (*Boström, Jackson & Simenstad, 2006*). *Terlizzi et al. (2010)* reported that 98% of the changes in benthic faunal assemblages between control and fish farm impacted sites were the result of changes in sediment features (sediment organic matter, grain size) and seagrass bed structure (shoot density, rhizome matte compactness). As an important indicator species, the abundance and biomass of *C. capitata* was most correlated to epiphytic suspension-feeding hydroid cover (%H) as well as biogeochemical sediment conditions (sediment organic content, microphytobenthos). Increases in epiphyte loads, both annual algae and suspension feeders, as well as annual benthic macroalgae and microphytobenthos have been associated with increased organic and nutrient loading from fish farms or other sources (Table 1; *Lever & Valiela, 2005*; *Ben Brahim et al., 2014*), thus reflecting enrichment conditions.

Our assessment of the correlation between macroinfaunal composition and environmental factors may be confounded by the fact that the fish farm was not in operation at the time of our sampling in July 2015. Finfish aquaculture operations are known to discharge significant quantities of nutrient and organic waste to the surrounding

environment. When in operation, the Port Mouton Bay fish farm increased the annual total dissolved nitrogen load from human sources to the entire bay by 14% or 30,400 kg ((*McIver et al., 2018*), with the fish farm being the single largest contributor of dissolved nitrogen to the bay after atmospheric deposition. The release of particulate organic waste from fish farms can be significantly larger (10s–100s mt) depending on the species raised, the scale of production and feed conversion efficiencies (*Olsen, Holmer & Olsen, 2008*). Impacts of fish farms on surrounding ecosystems can persist for several months to years after production ceases (*Delgado et al., 1999*; *Karakassis et al., 1999*; *Brooks, Stierns & Backman, 2004*; *Pereira et al., 2004*; *Lin & Beiley-Brock, 2008*). As no waste was being discharged from the farm site during our sampling season, our results likely represent the long-term cumulative impacts of organic loading from the fish farm rather than a direct immediate effect. Continued monitoring at the Spectacle Island and Carters Beach sites is highly recommended to assess possible eelgrass habitat recovery and to provide managers with scientific information on the development of monitoring metrics and impact thresholds that would protect eelgrass from the adverse effects of finfish aquaculture.

## Monitoring metrics for managing impacts

The sensitivity of seagrass habitats to environmental change and their link to sediments and the water column make them ideal bioindicators of a wide range of anthropogenic stressors (e.g., shading, shoreline modification, nutrient and organic loading, climate change, ocean acidification) in coastal marine environments (*Oliva et al., 2012*; *Roca et al., 2016*). Consequently, many multi-metric indices and management tools incorporating seagrass physiological, individual, population and community-level traits have been developed to aid in monitoring general trends in ecosystem status, assessing environmental quality, and evaluating impacts of development projects (*Romero et al., 2007*; *Oliva et al., 2012*; *García-Marín et al., 2013*; *Roca et al., 2016*). An advantage to using multiple metrics is that some measures, such as tissue %N and $\delta^{15}$N, serve as an early warning because seagrasses respond relatively quickly (days to weeks) to nutrient disturbances while population-level measures, such as shoot density or biomass, reflect the impact of disturbances on longer time scales (months to years) (*Romero et al., 2007*; *Roca et al., 2016*). A single indicator does not allow for spatial and temporal (seasonal) fluctuations and potential sampling errors, account for different levels of organizational response (from physiological to community) or capture the lag-time in response of some metrics to either the degradation or recovery phase (*Oliva et al., 2012*).

Unlike *Z. marina* in Atlantic Canada, the impacts of netpen finfish farming on *P. oceanica* in the Mediterranean have been the subject of extensive research (Table 1) with the goal of identifying monitoring metrics and management tools (*Holmer et al., 2008*; *Karakassis et al., 2013*). As a result of these studies, specific recommendations, such as setting monitoring thresholds (e.g., >1.5 g organic matter m$^{-2}$ day$^{-2}$ results in seagrass decline; (*Díaz-Almela et al., 2008*) and siting restrictions (e.g., no fish farms <800 m from the edge of seagrass beds; (*Karakassis et al., 2013*) have been made to improve the regulatory schemes for protecting *P. oceanica* beds from the impacts of finfish farms. While there are some physiological (e.g., tolerance to sulphides; (*Hasler-Sheetal & Holmer, 2015*)

and morphological (e.g., rhizome diameter; *Marbà & Duarte, 1998*) differences between seagrass species, the metrics identified by research done elsewhere could be applied to eelgrass conservation and management in Atlantic Canada. Currently, there are no specific monitoring thresholds or management tools in place to protect eelgrass from the potential impacts of finfish aquaculture in Atlantic Canada. Based on results from our study, from the Mediterranean (Table 1) and the above discussion, we recommend the development and application of the following metrics: sedimentation rates; sediment sulphide and organic content; microphytobenthos concentration; prevalence of *C. capitata*; epiphyte load and composition; eelgrass bed structure (shoot density, percent cover, above- and belowground biomass); and tissue nitrogen content and isotopic signature. Our selection of indicators is consistent with the key indicators identified for the response and recovery of seagrasses to nutrient and organic loading in a recent global review and assessment (*Roca et al., 2016*). As for siting fish farms near eelgrass habitat, we recommend that more research needs to be done to define appropriate siting and zoning criteria for open netpen fish farms (e.g., *Vandermeulen, 2005*). However, based on our study results and research done elsewhere, we recommend a precautionary setback of 1,000 m between eelgrass beds and open netpen finfish farm to ensure the conservation and protection of this sensitive and ecologically important habitat.

## Conclusions

Within the last five to ten years, there have been growing concerns about the potential significant spread and persistence of finfish aquaculture waste over large spatial areas (*Price et al., 2015*). Our results reveal changes to eelgrass bed structure and associated macroinfauna communities within Port Mouton Bay, particularly in closer proximity (<1 km) from the fish farm site. Given the acknowledged status of eelgrass as an Ecologically Significant Species in Atlantic Canada (*DFO, 2009a*), its key role in supporting productive and diverse species communities (*Larkum, Orth & Duarte, 2006*), and its sensitivity to anthropogenic disturbances, including finfish aquaculture (*Vandermeulen, 2005*; *DFO, 2009a*; *DFO, 2012*), management needs to include comprehensive eelgrass habitat assessments and zoning criteria as part of the process for evaluating proposed finfish aquaculture operations. Effective indicators, with threshold values, need to be developed to monitor existing operations to ensure that fish farms are operating in a sustainable manner. Optimal indicators should integrate impacts over time and include multiple measures of eelgrass canopy structure, tissue and environmental variables rather than single-event and individual variables.

## ACKNOWLEDGEMENTS

We thank K. Wilson, A. Dixon and T. Harington for support during the field surveys, A Chan for help in the lab, and M. Wong, M. Skinner, J. Grant, and S. Courtenay for helpful comments and discussions. We would also like to thank the Friends of Port Mouton Bay Society for providing their historical and contemporary knowledge of the Port Mouton area and marine ecosystem.

### Funding
Financial support was provided by the Natural Sciences and Engineering Research Council of Canada with a grant to Heike K. Lotze (NSERC RGPIN-2014-04491), and the OceanCanada Partnership (Grant # 22R20430). The funders had no role in study design, data collection and analysis, decision to publish, or preparation of the manuscript.

### Grant Disclosures
The following grant information was disclosed by the authors:
Natural Sciences and Engineering Research Council of Canada: NSERC RGPIN-2014-04491.
OceanCanada Partnership: # 22R20430.

### Competing Interests
The authors declare there are no competing interests.

### Author Contributions
- Nakia Cullain conceived and designed the experiments, performed the experiments, analyzed the data, prepared figures and/or tables, authored or reviewed drafts of the paper, approved the final draft, wrote the paper.
- Reba McIver performed the experiments, analyzed the data, prepared figures and/or tables, authored or reviewed drafts of the paper.
- Allison L. Schmidt conceived and designed the experiments, performed the experiments, analyzed the data, contributed reagents/materials/analysis tools, authored or reviewed drafts of the paper, approved the final draft.
- Inka Milewski contributed reagents/materials/analysis tools, authored or reviewed drafts of the paper.
- Heike K. Lotze performed the experiments, analyzed the data, contributed reagents/materials/analysis tools, prepared figures and/or tables, authored or reviewed drafts of the paper.

### Data Availability
   The raw data are provided in a Supplemental File.

### Supplemental Information
Supplemental information for this article can be found online at http://dx.doi.org/10.7717/peerj.5630#supplemental-information.

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
