# Peer review of "Potential impacts of finfish aquaculture on eelgrass (Zostera marina) beds and possible monitoring metrics for management: a case study in Atlantic Canada"

_PeerJ, doi:10.7717/peerj.5630_

## Round 0.1 · original submission · Major Revisions

Both of the reviewers and I all agree that the present work merits publication. The paper is generally well written, however needs major revision before its acceptance.

It is necessary to improve methods (see Rev 1 suggestions),
To complete Table 1.,To justify the interpretation on the non significante results and explain your findings according to Rev 1 concerns. After you have followed both Rev suggestions, we will really appreciate to receive it again for its consideration

Reviewer 1 ·

Basic reporting

The paper is geneally well-written and easy to read and follow – there are few typos, which I have pointed out in the last section of the review, but these can be easily fixed.
I do have issues about the structure of the manuscript in its present form – in particular there is information that is missing in the methods:
-notably, a lack of information about the 9 additional sites
-and details about the sites that are brought up only in the discussion, but should be moved to the methods section.

In addition, I was surprised to find that discussion jumped so quickly into the use of metrics. The title, abstract, introduction, methods, and results are very focused on the present study, but the discussion mostly focuses on using metrics for determining impacts. This needs to be better balanced throughout the manuscript (starting with changing the title to indicate this)! I LIKE that the authors are discussing how these metrics could be used, but this should be clear.

I would also like the 9 additional study sites to also be listed in Table 1, and shown on the map in Figure 1. I wonder also if it would be appropriate to add this data to the raw data, as there it only shows your four sites.

There are also legends missing from some graphs (I point this out below).

Experimental design

The experimental design is ok. It would be nice to have more than one control site, but you do make up for this by using the 9 additional sites (and I like that you pulled in additional data to fill the gap).

As I mentioned above, the manuscript addresses two important but different questions: First the impact of this particular fish farm, and secondly the use of metrics, but the title, intro, and abstract do not accurately reflect this.

I would like it to be pointed out explicitly in the methods that the fish farm did not actually have any fish in it when you did your sampling. This is stated, but it seems a bit hidden in the methods.

I also have a suggestion for the stable isotope analysis: you have the delta C and delta N of your eelgrass in all the sites, and you seem to also known the values of the potential sources (lines 354-358 and 370-373). Have you considered using a diet-mixing analysis which could possibly tell you where the nitrogen and carbon is coming from?

Validity of the findings

I have a major issue with how you are interpreting your non-significant results:
In lines 243-245, you say that there was lower shoot density, but at the same time say that it is non-significant. You do the same in lines 255-257 – either the results are lower/higher or they are non-significant, you cannot state both!

Again in the discussion (lines 297-299, line 384, lines 417-419) you say that you found impacts on shoot density, cover, biomass, and then on macroinfauna community metrics – but the only significant finding in this is the eelgrass biomass! You are severely misrepresenting your results here!

That said, I cannot quite interpret the figures in line with your statistical results – based on the error bars, it seems there should be significant differences between: SI and OW in Figure 3a, and then CB and the other sites in Figure 3b, CB and the other sites in Figure 4a, and between PJ and the other sites in Figure 4b.

I suggest you have another look at your statistical approach, and then properly report your results (and write your discussion) based on significance.

Additional comments

Note: some of this may repeat what I've said above, but I hopefully it is valuable to see it line-by-line

Title: As previously mentioned, change the title to give some idea that this paper is about the use of metrics for analysing impacts

Abstract: As with the title, revise the abstract to match the scope of the manuscript

Introduction:
Lines 32-33: citation for "most diverse of the soft-bottom marine communities"?
Lines 43-44: some expansion on the actual effects seen on seagrass meadows (increase/decrease, etc.)
Lines 51-53: Need some references for this
Lines 66-67: Expand on these potential impacts
Lines 75-77: Expand on this section to match the discussion

Methods:
Line 96: Explicitly state here that at the time of sampling there were no fish.
Lines 99-101: State the distances of each site here
Line 111: How far apart were the transects?
Lines 125-126 and 128: This is confusing, you state several taxa and then mention only hydroids? If you are only using hydroids, then remove the references to the others.
Lines 166-175: Move this section to the beginning of the Methods section or integrate with the last paragraph of the introduction.
Lines 174-175: Expand upon these 9 sites. Where are they (add them to the map)? And how are they used statistically? Are you taking the mean of all of them and treating them as a fifth site?

Results
Lines 242-245: These are not significant, do not state them as if they are
Line 255-257: Not even close to significant (according to the p-values above), there is no trend.
Line 260: Fix typo: “no overlap” to “noT overlap”
Line 262: change “lied” to “lay”
Line 278: “also” should not be capitalised
Line 280: remove “beach”

Discussion
Line 297: The only significant result is eelgrass biomass!
Line 298: Also not significant!
Line 302: Name the site
Lines 306-307: As I mentioned above, you need to better integrate these two goals (impacts in your site AND using metrics) into your whole manuscript. I would also suggest that based on your non-significant results, eelgrass cover and density are NOT appropriate metrics, not is infaunal biomass and abundance.
Lines 310-334: Most of the this section should be moved into site description in the methods (except for lines 326-328 abd kubes 332-334)
Lines 344-346: A sentence or two of the ecological consequences of anoxia/hypoxia would be good here.
Lines 354-357: As I mentioned above, it seems you would have the data to do a nice diet-mixing analysis here!
Lines 366-369: This is good! I am wondering if the cold event could have had an effect on the eelgrass community also?
Lines 384-385: NOT significant, exept biomass!
Line 386: The “TWO” closest sites
Line 416-417: once again, you cannot state “our results indicated lower biomass” especially as your say right before that there were no significant differences

Table 1
-“Wharf” is misspelled
-Add the 9 additional sites to this table (or at least their mean and SE)
-What exactly is the difference between muddy sand and sandy mud?
-Highlight that PJ is the control site

Table 3: Change the last sentence to : “Species identified in SIMPER analysis”

Table 4: Same as Table 3

Figure 1:
-Add the 9 additional sites to a map of Nova Scotia.
-Remove the “disappeared” sites

Figures 2, 3, and 4:
Note in the captions that the distance from the fish farm increases from left to right, this makes it much easier to interpret the results. Also, please add the colour legend to the actual graph, it’s much easier to read that way.

Figure 2
2a: Change to “Sediment Organic CONTENT”
2e: Flip the axis to make it consistent with the others graphs (just note it in the caption)

I pointed these out above, but the error bars in Figures 3 and 4 do not match the stats:
3a: Based on the error bars, SI and OW should at least be significantly different?
3b: It looks like CB should be different?
4a: Same here - looks like CB should again be different?
4b: Here it looks like PJ should be different
4c: OW and PJ look different

Figure 5.
There is missing a legend here: Which curve is the abundance and which is the biomass?

Reviewer 2 ·

Basic reporting

This MS reports results from a study in which the authors have looked at how finfish farming affects eelgrass beds and associated infauna. They show that the effect of farming is obvious on some of the plant and faunal variables. The language is clear and professional throughout the MS. The introduction is straight-forward and gives a good background to the study with relevant literature referenced. The structure conforms to the journal’s standards. The figures are for the most part well labelled, though a few things could be improved (see my further comments for the authors). The raw data is also provided and structured coherently.

Experimental design

The research is original for this particular study organism (Zostera marina). The research aims are defined and relevant, though to clarify I would like the authors to give some examples of hypotheses (see my comments). The methods are described with sufficient detail and can be replicated. Some details can be clarified (see my comments).

Validity of the findings

The data is robust and the authors have used proper statistical methods in the MS. The conclusions are for the most part well-stated but I would like the authors to clarify and justify some of their statements based on the results (see comments). I believe that the authors could speculate more instead of stating facts that are not necessarily supported by their results.

Additional comments

I think the MS is generally well-written with concise methodology, good reporting of the results and discussion, and thus adheres well to the standards of Peer J. However, there are some issues that I think should be clarified and improved before acceptance. Please see my additional comments.

Annotated reviews are not available for download in order to protect the identity of reviewers who chose to remain anonymous.

---

## Round 0.2 · Minor Revisions

The Reviewer and myself consider you have done an excellent job addressing the concerns of the original submission. The submission is as good as accepted, but please follow carefully the suggestions made for the reviewer such us some English corrections and some methods clarifications.

Reviewer 1 ·

Basic reporting

The authors have done an excellent job addressing the gaps that I pointed out in the original submission. They have added thorough information about the other sites across Nova Scotia, and the modified title and abstract much better reflect the scope of the paper.
One minor suggestion on this point would be to explicityly state the development of metrics as an objective of the paper in Lines 78-90.
I also really appreciate the added information in Table 1! This really adds puts the results of this study into a wider context.

A few minor corrections to make in the text and figures:
Line 21: change “identify” to past tense “identified” to match the previous section
Line 99: change “4” to “four”
Line 131: change “date” to “data”
Line 506: change “above 192” to “>192”
Figure legends: In Figures 2-5, there is a sentence stating “pairs of the same letter in bold would be further different a p<0.01”. First, I think there is a typo here and it should be “at” instead of “a”? Secondly, I’m not quite sure what this actually means – in Figure 5b, the highest ‘b’ and the lowest ‘b’ are bold, but not the one in the middle. If there are ‘marginally significant’ (p<0.1) values perhaps they should just be mentioned in the legend rather than confusing the figure text.
Figure 6: Maybe add the site abbreviation on the graph panels. This makes it easier to read.
Figure 7: What is the “distance” in panel C referring to?
Table 4: I think this would benefit from being turned into a Figure, especially for the important species that are discussed in lines 490-510.

Experimental design

As mentioned above, the authors did an excellent job adding information about the additional sites and re-analysing the data.

Minor improvements:
Line 115: Do you have anything to put this 4000 uM number into context? What is considered high or low for this?
Line 147: Add which years the other sites were sampled (these could be also added in Table 2).

Validity of the findings

The authors have softened some of the conclusions, and instead separate “statistically significant” and “ecologically relevant” results. This is an improvement, though I still think it needs to be made clearer which results are more statistically supported versus which show a potential trend and need to be followed up on with larger scale studies of different fish farms (this last point is not a criticism of the paper – I think this manuscript is an excellent first step towards quantifying aquaculture effects on eelgrass!)

In line 355 (and line 23 in the abstract), the authors mention that there is higher algae % cover is an effect of the fish farm. However, when looking at Figure 2c, I completely disagree - only site CB is higher than the Nova Scotia average and the control site. The other two sites in the bay with the fish farm are the same or even lower than the control site. Either remove this completely, or discuss why there might be an algal biomass increase at interediate distance from the fish farm, but not closer or further away. From the graph in Figure 2c, it looks like algal biomass is random and not linked to the fish farm at all.

Additional comments

The authors did excellent work on improving the manuscript. There are a few areas in the discussion that I think could still be improved (in terms of discussing significant and non-significant results), and a few minor points to modify, all of which I have outlined above.

---

## Round 0.3 · accepted · Accept

Having reviewed the final changes made, I inform you that your paper is ready for publication.

#